# In Differential Privacy, There is Truth:
# On Vote Leakage in Ensemble Private Learning

**Jiaqi Wang**[a b], **Roei Schuster**[b], **Ilia Shumailov**[b c], **David Lie**[a], **Nicolas Papernot**[a b]

[a]University of Toronto
[b]Vector Institute
[c]University of Oxford

## Abstract

When learning from sensitive data, care must be taken to ensure that training algorithms address privacy concerns. The canonical Private Aggregation of Teacher Ensembles, or PATE, computes output labels by aggregating the predictions of a (possibly distributed) collection of teacher models via a voting mechanism. The mechanism adds noise to attain a differential privacy guarantee with respect to the teachers' training data. In this work, we observe that this use of noise, which makes PATE predictions stochastic, enables new forms of leakage of sensitive information. For a given input, our adversary exploits this stochasticity to extract high-fidelity histograms of the votes submitted by the underlying teachers. From these histograms, the adversary can learn sensitive attributes of the input such as race, gender, or age. Although this attack does not directly violate the differential privacy guarantee, it clearly violates privacy norms and expectations, and would not be possible *at all* without the noise inserted to obtain differential privacy. In fact, counter-intuitively, the attack *becomes easier as we add more noise* to provide stronger differential privacy. We hope this encourages future work to consider privacy holistically rather than treat differential privacy as a panacea.

## 1 Introduction

The canonical Private Aggregation of Teacher Ensembles, PATE, is a model-agnostic approach to obtaining differential privacy guarantees for the training data of ML models [1], [2], that is widely applied [3], [4] and adapted [5]–[7] due to its comparatively favorable trade-off between differential privacy, utility, and ease of decentralization [6]. In PATE, one considers an ensemble of independently trained teacher models. To generate a prediction, PATE first collects the predictions of these teachers to form a *histogram* of votes. It then adds Gaussian noise to the histogram and only reveals the label achieving plurality. This label can be used directly as a prediction, or to supervise the training of a student model—in a form of knowledge transfer. Because PATE only reveals the label receiving the most votes, it comes with guarantees of differential privacy, i.e., the noisy voting mechanism allows us to bound how much information from the training data is potentially exposed [8].

But PATE does not explicitly protect from leakage of a key element in its inference procedure: the histogram of votes submitted by teachers. While the histogram is used internally and not directly exposed to clients, a careful examination of PATE reveals that information about the histogram leaks to clients via query answers.

The histogram can contain highly sensitive information, not the least of which is membership in minority groups which, if revealed, can be used to discriminate against individuals. We demonstrate this by showing how an attacker, using the vote histogram of a PATE ensemble trained to predict an individual's income, can infer wholly different attributes such as their level of education, even when the attacker's instance does not contain any information related to education-level. Why is this

36th Conference on Neural Information Processing Systems (NeurIPS 2022).

possible? At a high level, the histogram of votes can be interpreted as a relatively rich representation of the instance, that reveals attributes beyond what the ensemble was designed to predict.

Next, we ask, is this attack a realistic threat? We answer this in the affirmative by designing an attack that extracts PATE histograms by repeatedly querying PATE, and showing that it reconstructs internal histograms to near perfection. Our attack builds on the fact that repeated executions of the same query produce the same internal histogram and a consistent distribution of PATE's noised answers corresponding to this histogram. Our adversary can thus sample this distribution many times via querying, and use it to reconstruct the histogram.

This implies that our attack relies on the stochasticity of PATE's output, which is a product of Gaussian noise, the very mechanism that was intended to protect privacy. In fact, we find that the larger the variance of noise added to the histogram votes, the more successful our adversary is in reconstructing the histogram. This is in sharp contrast with the known and expected effect in differential privacy, that higher noise scale generally leads to stronger privacy. Put simply: differential privacy makes our attack possible.

An astute reader may observe that histogram leakage does not violate the differential privacy guarantee, which only protects individual users in the training data, which is not compromised here. While it is absolutely true that our attack does not violate differential privacy, it clearly violates societal norms and user expectations that differential privacy is often incorrectly assumed to protect. The fact that differential privacy enables the leakage we exploit nicely underscores the distinction between technical definitions of privacy and common conceptions of privacy.

The attack is difficult to mitigate. Particularly, we show that it is stealthy in the sense that PATE's own accounting of "privacy cost" considers our attacker's set of queries "cheap", meaning that revealing their answers has a relatively small effect in terms of differential privacy. Consequently, PATE's privacy-spending monitoring does not prevent our attack. Our attack also performs only a moderate number of queries in absolute numbers, the same number used by common legitimate PATE clients, so a hard limit on queries would impede PATE's utility. We will discuss other mitigation approaches, which are not robust and/or not always usable.

To summarize, our contributions are as follows:

- We posit the novel threat of extracting PATE's internal vote histograms. We observe and show that those contain sensitive information such as minority-group membership.
- We show that differential privacy is the cause for histogram-information leakage to PATE's querying clients.
- We exploit this leakage to reconstruct the vote histogram. We achieve this by minimizing the difference between (a) the probability distribution of outcomes observed by repeatedly querying PATE and (b) an analytical counterpart that we derive.
- We experiment with standard PATE benchmarks, showing that the attack can recover high-fidelity histograms while using a low number of queries that remain well within PATE's budget intended to control leakage.

## 2  Vote Histograms are Sensitive Information

We consider an ensemble's vote histogram, such as those computed internally in PATE. Clearly, such histograms contain a lot more information on PATE's innerworkings than simply its revealed decision, but it is important to clarify that there are common contexts in which this leakage can actually be used to hurt individuals as they contain sensitive information about them.

As a prominent example, minority-group membership often leaks via histograms, and can of course be used to discriminate against group members. To understand this, let's consider a minority group that is under-represented in the training data distributed across PATE's teachers. Each teacher observes some outliers and mis-representitive phenomena such as coincidental correlations or out-of-distribution examples. When data on group members is scarce, each model will tend to over-fit to the outlier phenomena within its own data, creating inter-model inconsistencies and resulting in disagreement, or low consensus, when predicting on similar inputs at test time—which readily presents itself on vote histograms. Thus, *we expect histograms to reveal members of minority group members via low consensus values*. Next, we illustrate this via a simple experiment.

**Extracting sensitive attributes from UCI Adult-Income histograms.** We now simulate an attack that receives the vote histogram of a salary-predictor ensemble and uses it to detect a small minority of the population, specifically, PhD holders. Following the above observation, our attack will simply classify all highly-consensus (consensus > 75%) predictions as non-PhD-holders, whereas low-consensus (< 75%) predictions will be classified as PhD holders. This is a heuristic attack that relies on intuition rather than learning the ensemble's behavior using a labeled dataset. On one hand, it may underestimate the attacker's ability to detect PhD holders; on the other hand, it does not require a labeled dataset and only assumes that the attacker sees the votes histogram.

We use UCI Adult-Income dataset [9], containing around 41,000 data points with basic personal information on people such as age, work hours, weight, education, marital status, and more. PhD holders form about 1% of this dataset. We randomly selected 80% of the dataset for training, and held out the rest for testing. We randomly partitioned the data into 250 disjoint sets. For each, we fitted a random forest model (using a hyperparameter grid search, see Appendix D) predicting whether income is above or below $50,000. For both training and test data, we removed the data columns explicitly indicating education levels, that is, training and test individuals do not contain any feature that directly distinguishes PhD from non-PhD holders.

Figure 1 shows the distribution of high-consensus and low-consensus on the test set (to make the effect clearer, we balanced the minority and majority groups in the test set by randomly removing most of the non-PhD samples). We observe that low consensus indeed indicates minority-group membership. Our attacker's precision is not particularly high (75% on the balanced set), but they can still use this signal to discriminate against minority groups.

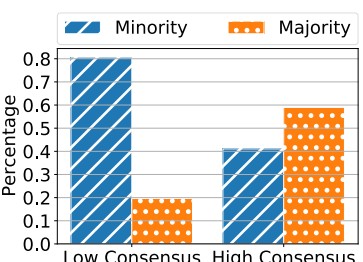

Figure 1: High vs. low-consensus distributions of the PhD-detection attack: vote histograms of minority-group members present lower consensus, allowing an attacker to identify them.

**End-to-end scenario and other attacks.** Appendix E presents this attack in an end-to-end scenario where the attacker does not have direct access to the histogram, and has to first query a PATE instance to infer it, using our methodology in Section 3. More sophisticated attackers can look for distinctive histogram patterns that characterize certain groups; the attack should become more accurate as more models are added to the ensemble, refining the attacker's histogram measurement; and precision can be amplified if the attacker holds multiple samples that are known to belong to the same group. Further, we note that sensitive-attribute extraction is not the only example for when vote histograms leak sensitive information: an attack could use votes to try to infer dataset properties [10] or distinguish between different partitions of the data associated with the different teachers in the ensemble.

## 3 How to Extract PATE Histograms

Having established that vote histogram leakage poses a risk to privacy and fairness, we proceed to provide a generic method for extracting vote histograms from PATE.

### 3.1 Problem Formulation and Attack Model

**A primer on PATE.** The PATE framework begins by independently training an ensemble of models, called teachers, on partitions of the private data. There is no particular requirement for the training procedure of each of these teacher models; the only constraint is that the partitions be disjoint. Queries made by clients are answered as follows: (1) each teacher model predicts a label on the instance, (2) the PATE aggregator builds a histogram of class votes submitted by teachers, (3) Gaussian noise is added to this histogram, and (4) the client receives the noised histogram's argmax class (henceforth *result class*). This noisy voting mechanism gives PATE its differential privacy (DP) guarantee, in what is an application of the Gaussian mechanism [11].

To preserve differential privacy, PATE tracks the *privacy cost* of the set of past queries, and stops answering queries once the cost surpasses the *privacy budget*. The cost computation is parameterized by a size $\delta$. The key differential privacy guarantee of PATE can be stated as follows: for a given set

of queries with cost $\varepsilon$, PATE is $(\varepsilon, \delta)$ differentially private. Put succinctly, $\varepsilon$ bounds an adversary's ability to distinguish between any adjacent training datasets, whereas $\delta$ bounds the (usually small) probability, over PATE's randomness, of this bound not holding. We defer additional details on differential privacy in PATE to Appendix A.

**Attacker's motivation.** Our attacker's goal is to recover the histogram of the vote counts when PATE labels an instance. Formally, given $N$ predictors $\{P_1, ..., P_N\}$ and a target input $a$, our attacker wants to infer $H \equiv Count(P_1(a), \ldots, P_N(a)) = [h_1, \ldots, h_c]$ where $Count$ counts the number of appearances of each element in $[c]$. Vote histograms can be used to extract potentially-sensitive information about an instance, such as its race, gender, or religion (see Section 2).

**Attacker's access and knowledge.** Our attacker can send queries to the aggregator and receive the label predicted by PATE (i.e., the output of the noisy voting mechanism). This may be possible because the aggregator willfully exposes the predictions of PATE, e.g., through a MLaaS API. Alternatively, fully-decentralized implementations of PATE have been proposed where the central aggregator is replaced with a cryptographic multi-party computation protocol [6], and its output is exposed directly. Figure 2 visualizes the workflow of our attack.

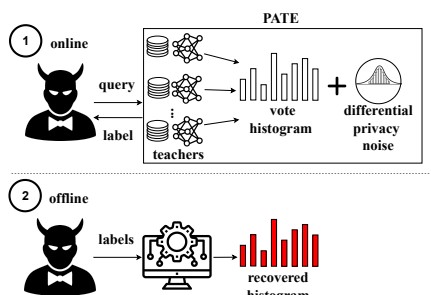

Figure 2: In an online phase, the attacker sends a specific query to PATE repeatedly and receives labels output by the noisy argmax. Offline, the attacker uses the labels to recover the histogram by constructing and solving an optimization problem.

In PATE, the parameters (mean, variance) of the noise added during aggregation are public domain [2]; we therefore assume the attacker knows them. We also assume the attacker knows the number of teacher models $N$, which may or may not be public. This assumption is only necessary to shift the attacker's learned distribution by a constant to attain a low $L_1$ approximation error when reconstructing histograms (Section 3). We note that the attacker could just as easily exploit the leakage (e.g. to learn sensitive attributes or differentiate between training sets) without it but we chose to instead make this assumption to simplify result presentation and interpretation.[1]

### 3.2 Our histogram reconstruction attack

The idea behind the attack, given in pseudo-code in Algorithm 2, is as follows: let $Q$ be a function that computes output-class probability distribution of PATE given a vote histogram $H$. First, our attacker will sample PATE to find an estimate for this distribution $\bar{q} \approx Q(H)$. Second, the attacker will use gradient descent to find $\hat{H}$ that minimizes the Euclidean distance between $Q(\hat{H})$ and $\bar{q}$. Finally, they shift the estimated histogram $\hat{H}$ by a constant to account for the number of teachers (this step assumes the number of teachers is known, but is done mostly for presentation purposes, see Section 3.1). We now detail these 3 steps.

**Step 1: Monte Carlo approximation.** The first step will sample PATE $M$ times and estimate the distribution over PATE's outputs $\bar{q} \approx Q(H)$ by setting each class probability as its Monte Carlo estimated mean frequency, i.e. $\bar{q}_i \leftarrow \frac{1}{M} \sum_{j=1}^{M} q_i^j$ where $q_i^j$ indicates whether class $i$ was sampled in the $j$th step. By the law of large numbers, as $M$ increases, $\bar{q}_i$ converges to $i$'s sampling probability $[Q(H)]_i$, and we can expect the attacker's estimate produced in the next steps to be more accurate. Our attacker would want to increase $M$ as much as possible, until they exceed PATE's privacy budget.

Indeed, in PATE, the privacy leakage expended by each individual query can then be composed over multiple queries to obtain the total privacy cost $\varepsilon$ needed to answer the set of queries. Once the total

---

[1]Indeed, we could avoid this assumption while still retaining low error if we measured the attacker's error with shift-invariant distances, like Pearson correlation.

privacy cost $\varepsilon$ exceeds a maximum tolerable privacy budget, PATE must stop answering queries to preserve differential privacy. Section 4 shows that the attack succeeds for values of $M$ that remain well below PATE's privacy budget, and are also moderate in absolute value, as they are similar to the query number of student models that use PATE.

**Step 2: constructing the optimization objective.** Our attacker wants to find $\hat{H}$ such that $\left\|Q(\hat{H}) - \bar{q}\right\|_2$ is minimized where $\|\cdot\|_2$ denotes the Euclidean norm. Given a (differentiable) closed-form expression for $Q$, it becomes natural to program and solve this with modern gradient-based optimization frameworks. Theorem 1 provides a closed form expression; and our attacker will use a differentiable approximation of this expression, as explained below.

**Theorem 1.** *Let $H = [H_1, \ldots, H_c]$ be the vote histogram for the $c$ classes, and let PATE's Gaussian-mechanism function $Agg(H) \equiv \operatorname{argmax}\{H_i + \mathcal{S}_i\}$ where $\mathcal{S} = [\mathcal{S}_1, \ldots, \mathcal{S}_M]$ is a vector of $M$ samples from a zero-mean normal distribution with variance $\sigma^2$. Then the probability that the randomized aggregator outputs the class $k$ is given by $[Q(H)]_k = \mathbb{P}(Agg(H) = k) = \int_{-\infty}^{\infty} \prod_{i=1}^{c,i\neq k} \Phi_i(\alpha)\phi_k(\alpha)\mathrm{d}\alpha$ where $\Phi_i(\cdot)$ is the cumulative probability distribution (CDF) of $\mathcal{N}(H_i, \sigma^2)$ (normal distribution with mean $H_i$ and variance $\sigma$) and $\phi_k(\cdot)$ is the probability density function (PDF) of $\mathcal{N}(H_k, \sigma^2)$.*

*Proof.* $Q(H) = \mathbb{P}(Agg(H) = k)$, is the probability that $H_k + S_k = \max\{Agg(H)\}$. For any $k$, $H_k + \mathcal{S}_k$ is a random variable that follows a normal distribution with mean equal to $H_k$ and variance equal to $\sigma^2$. Let $g_k = H_k + \mathcal{S}_k$, then $g_k \sim \mathcal{N}(H_k, \sigma^2)$. $[Q(H)]_k$ is the probability of $g_k$ is greater than $g_j, \forall j \in \{1, \ldots k-1, k+1, \ldots c\}$

$$
\begin{aligned}
[Q(H)]_k &= \mathbb{P}(Agg(H) = k) \\
&= \mathbb{P}(g_k > g_1, \ldots, g_k > g_{k-1}, g_k > g_{k+1}, \ldots, g_k > g_c) \\
&= \int_{-\infty}^{\infty} \prod_{i=1}^{c,i\neq k} \mathbb{P}(g_i < \alpha \mid g_k = \alpha)\,\mathbb{P}(g_k = \alpha)\mathrm{d}\alpha \\
&= \int_{-\infty}^{\infty} \prod_{i=1}^{c,i\neq k} \Phi_i(\alpha)\phi_k(\alpha)\mathrm{d}\alpha
\end{aligned}
$$

$\square$

---

**Algorithm 2** Attack pseudocode

**Input:**
1: $N \in \mathbb{N}$         ▷ total number of teachers (see Section 3.1 for why this is needed)
2: $\mathbb{O}$             ▷ PATE instance
3: $T, \lambda$    ▷ optimization termination threshold and learning rate

**Output:** $\hat{H}$
4: $S \leftarrow sample(\mathbb{O}, M)$   ▷ sampling PATE $M$ times and storing into $S \in 1, \ldots, K^M$
5: **for** $i = 1, 2, \ldots, M$ **do**
6:     **for** $j = 1, 2, \ldots, c$ **do**
7:         $q_j^i = int(S^i == j)$    ▷ $q_j^i = 1$ if $S^i = j$, 0 otherwise
8:     **end for**
9: **end for**
10: $\bar{q} \leftarrow 0^c$   ▷ initialization of $\bar{q}$ with a 0 vector
11: **for** $i = 1, 2, \ldots, M$ **do**
12:     $\bar{q}[i] \leftarrow \frac{1}{M}\sum_{j=1}^{M} q_i^j$
13: **end for**
14: $\hat{H} \leftarrow 0^c$ ▷ initialization of $\hat{H}$, here we use an all-zero array of length $c$
15: **while** $\left\|Q(\hat{H}) - \bar{q}\right\|_2 > T$ **do**
16:     $\hat{H} \leftarrow \hat{H} - \lambda\nabla_{\hat{H}}\left\|Q(\hat{H}) - \bar{q}\right\|_2$
17: **end while**
18: $\hat{H} \leftarrow \hat{H} + [\frac{N - \sum\hat{H}}{c}]^c$ ▷ shift $\hat{H}$ to sum to $N$
    **return** $\hat{H}$

---

The expression in Theorem 1 is not usable in automatic differentiation and optimization frameworks; we therefore use an approximation of the integral by the trapezoid formula. We select points with higher probability and sum up their values to get an approximation of the integral with infinite bounds. Then we decide what values to select. In the integral $\int_{-\infty}^{\infty} \prod_{i=1}^{m,i\neq k} \Phi_i(\alpha)\phi_k(\alpha)\mathrm{d}\alpha$, $\alpha$ is the value of $g_k \sim \mathcal{N}(H_k, \sigma^2)$. Therefore $\alpha$ has the highest probability at $H_k$, and has the higher probability closer to $H_k$. More specifically, properties of the normal distribution give us that $\mu \pm 6*\sigma$ covers 99% of the values of Gaussian random variable $z \sim \mathcal{N}(\mu, \sigma)$. Therefore values of $\alpha$ between

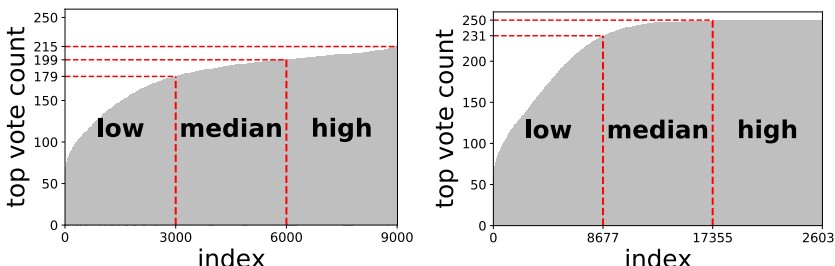

Figure 3: Divisions of the 9,000 and 26,032 histograms of MNIST (left) and SVHN (right) datasets into 3 consensus levels, measured by top-agreed label percentage. The dashed red lines delineate the 33.3% and 66.7% quantiles.

$H_k \pm 6 * \sigma$ cover 99% of the integral area. Therefore,

$$\int_{-\infty}^{\infty} \prod_{i=1}^{c, i \neq k} \Phi_i(\alpha)\phi_k(\alpha)\mathrm{d}\alpha \approx \sum_{H_k-6\sigma}^{H_k+6\sigma} \prod_{i=1}^{c, i \neq k} \Phi_i(\alpha)\phi_k(\alpha),$$

which is differentiable and is handled well by most automatic differentiation packages.

**Step 3: accounting for the number of teachers.** The distribution estimate produced by our optimization may be skewed by a constant because $[Q(H)]_k$ only depends on the differences between $g_k$ and $g_1, \ldots, g_{k-1}, g_{k+1}, \ldots, g_c$, so the attacker shifts each element of $\hat{H}$ by $(N - \sum \hat{H})/c$ so that the new histogram $\hat{H}$ sums up to $\sum \hat{H} + c * (N - \sum \hat{H})/c = N$. Theorem 2 in Appendix B provides proof that shifting $Q(\hat{H})$ by a constant does not affect $Q(\hat{H})$.

# 4 Evaluation

We evaluate our attack against instantiations of PATE on common benchmarks. We show that the extracted histograms only differ slightly from the true ones underlying PATE's decision. This is despite the low privacy cost of the attacker's queries, which remains well within budgets enforced by common PATE instantiations. We also quantify the impact of the choice of scale for the noise being added to preserve DP: we show that *higher noise values result in increased attack success* for a given number of queries. We offer an hypothesis to explain this ostensibly surprising observation.

## 4.1 Experimental Setup

**Data.** We use the experimental results from Papernot et al. [1] to simulate our attack environment. Papernot et al. released the histograms obtained by PATE using 250 teachers for two 10-class computer-vision benchmarks, MNIST [12] and SVHN [13]. There are 9,000 histograms generated by MNIST experiments and 26,032 histograms generated by SVHN experiments, corresponding to the sizes of these datasets' test sets.

We define a histogram's *consensus* as its maximum value, and divide each dataset into three equal-sized groups corresponding to high consensus, medium consensus, and low consensus. Figure 3 illustrates this. We sample five histograms randomly from each group, and mount our attack for various noise levels.

**Attack parameterization.** We simulated attackers with two types of query limits: first, an attacker limited by PATE's canonical privacy budget; we used the parameterization from Papernot et al. [1], i.e. budgets of 1.97 and 4.96 for MNIST and SVHN and $\sigma = 40$. Second, an attacker with a hard limit of $10^4$ queries; this is a moderate number of queries for clients wishing to train their own "student" model using the aggregator's labels (see [1], [2]). We applied this attack against PATE instantiations for MNIST and SVHN with noise levels $\sigma \in \{40, 60, 80, 100\}$.

For optimization (see Section 3), we use an adaptive learning rate: at the beginning of training, we use a learning rate of $\frac{10}{\left\| \nabla_{\hat{H}} J \right\|_2}$, where $J = Q(\hat{H}) - \bar{q}$ is the optimization objective. As the optimiza-

tion starts to converge, $\frac{10}{\left\|\nabla_{\hat{H}} J\right\|_2}$ becomes too large so we switch to a learning rate of $\frac{1}{\left\|\nabla_{\hat{H}} J\right\|_2}$. This results in changes to the histogram of the magnitude of one vote for each update. We use 0.01 as a threshold on the loss to establish convergence, and thus when $\|J\|_2 < 0.01$, we stop optimizing. For the attacks against canonical settings, we stopped once estimated histograms started presenting negative values, which we found to be a slightly better strategy. (We could also try to constrain it to only-positive values; we discuss improving this optimization procedure further in Section 5).

**Metrics.**  For every attack, we measured the *error rate* and *privacy cost*. The error rate is defined as the normalized $L_1$ distance between the ground-truth histogram $H = [H_1, \ldots, H_c]$ and our attacker's estimate $\hat{H} = [\hat{H}_1, \ldots, \hat{H}_c]$, i.e., $\sum_i \left| H_i - \hat{H}_i \right| / (2 \sum_i |H_i|)$. (While the optimization minimizes Euclidean distance, we report L1 errors because they can be interpreted as corresponding to the number of mis-counted votes.)

We define and compute the privacy cost incurred by the adversary using established practices. At a high level (see details in [1]), we model PATE as a Rényi-differentially-private mechanism and leverage known privacy-preserving-composition theorems; we attain (non-Rényi) differential privacy via a known reduction from differential privacy to Rényi differential privacy.

The parameter $\delta$ is set as $10^{-5}$ for MNIST and $10^{-6}$ for SVHN, following Papernot et al. [1].

**Implementation**  Our implementation is provided in Python and the optimization uses the Jax library.  Our code is open-sourced at `https://github.com/cleverhans-lab/monte-carlo-adv`.  We ran the optimization on an Intel Xeon Processor E5-2630 v4; it takes about 2.5 hours to complete for a single histogram.

### 4.2  Results

**Our attack has high performance within canonical privacy budgets.**  We first evaluate our attack on canonical PATE from Papernot et al. [1]. Figure 4a and 4b show our attacker's error rates for the different histograms, averaging 0.11 on the MNIST setup and 0.05 on the SVHN setup.

**Our attack extracts high-fidelity histograms and has low privacy costs.**  Figure 4 reports the performance of the privacy-budget limited attack; Figures 6 and 7 show our hard-query-limit attacker's error rate and query costs for different noise levels, i.e. values of $\sigma$. We observe that, across attacks, we attain very low error rates, often as low as 0.03, translating to 3% of the votes being miscounted. For the hard-query-limit attack, privacy costs roughly range between 1 to 12, which is the order of magnitude for the budget one would plausibly use, for example to attain guarantees similar to Papernot et al. [2] (which uses budgets of up to 8 in a directly comparable setting to ours) or Abadi et al. [14] (which also employs a $(8, 10^{-5})$-differentially private mechanism for MNIST).

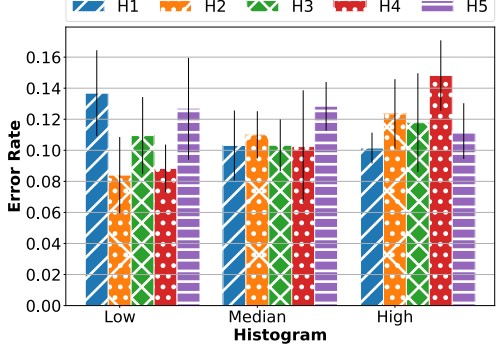

(a) Error rates on attacking a canonical MNIST PATE with privacy budget = 1.97 and $\sigma = 40$

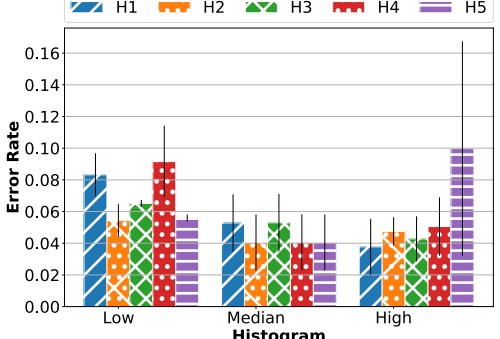

(b) Error rates on attacking a canonical SVHN PATE with privacy budget = 4.96 and $\sigma = 40$

Figure 4:  Error rates on budget-limited attack on the canonical PATE [1], for our 15 low/median/high-consensus sample histograms.

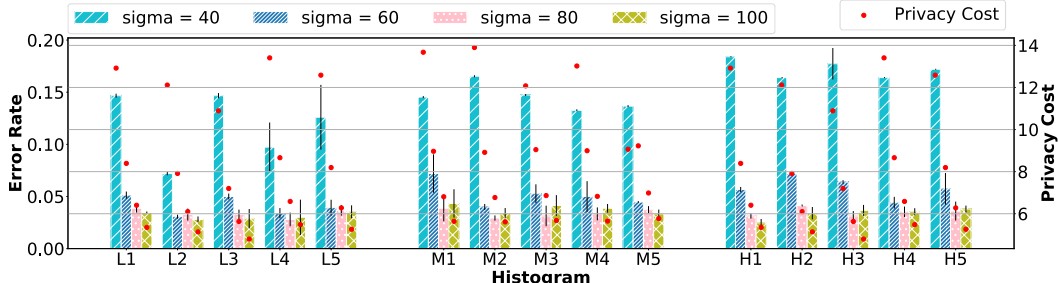

Figure 6: Our attack's error extracting 15 MNIST histograms with low/medium/high consensus (L1-5, M1-5, and H1-5 respectively) using different noise scales and a query limit of $10^4$. The red dots and the right axis show the privacy cost of the attack on each histogram.

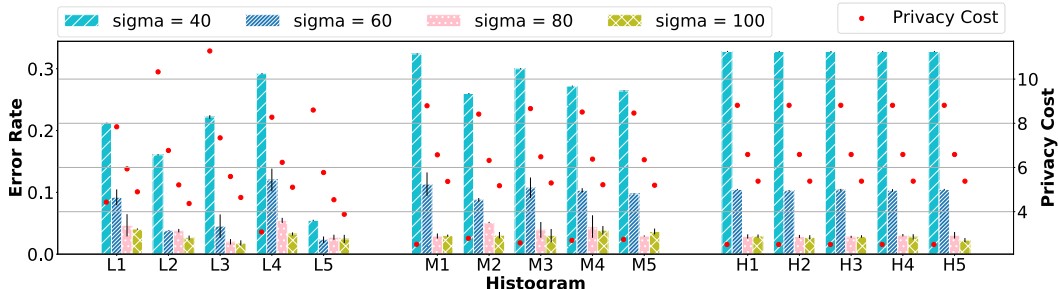

Figure 7: Our attack's error extracting 15 SVHN histograms with low/medium/high consensus (L1-5, M1-5, and H1-5 respectively) using different noise scales and a query limit of $10^4$. The red dots and the right axis show the privacy cost of the attack on each histogram.

**Adding more noise helps the attacker.** Perhaps the most surprising result in this work is that the higher the noise scale, the lower the attacker's error is. This is *not* necessarily aligned with using up more of the privacy budget. In fact, in many cases, increasing the noise decreases both the attacker's privacy cost and their error; Figure 5 shows the correlation between cost and error.

This is counter-intuitive, as larger Gaussian scales $\sigma$ usually correspond to tighter privacy guarantees. That is, more expected protection against attacks. Specifically, our Monte Carlo estimation should be less accurate when higher-variance noise is added, as convergence to the mean is slower. Nevertheless, our attack actually performs *better* with higher noise levels.

To explain this, consider the aggregator's output distribution. When it

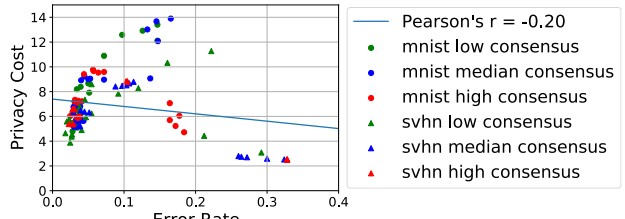

Figure 5: Our attack's average error rate vs. privacy cost on the histograms extracted with $10^4$ queries. Weak inverse correlation implies cheaper attacks are often more accurate.

is uniform, classes are sampled with equal probabilities, contributing equal information to each Monte Carlo estimator. Conversely, when some classes have a lower probability than others, their estimator will receive less samples. Sharp output distributions, for example, have a peak that essentially "eclipses" other classes. To illustrate this, consider the case where no noise is added at all; here, the output is always the plurality vote, and a black-box querying adversary cannot learn *anything* about the histogram except its top-voted class, which is already known after a single query.

Our results indicate that the mitigation of this eclipsing effect by increasing the noise, can be more dominant than the adverse effect that increasing noise has on Monte Carlo convergence. Interestingly, this is not always reflected in PATE's privacy-cost score, which is often lower for setups that leak more on vote histograms. Technically, there is no contradiction: privacy cost measures differential privacy, which does not necessarily translate to protection against vote-histogram leakage.

# 5  Discussion

**Mitigation.**  The possibility of this attack is inherent to PATE's aggregation mechanism, as long as the attacker can make multiple queries to PATE. Our experiments in Section 4 show that (1) using tighter privacy budgets does not necessarily mitigate the attack, as there is no strong correspondence between the privacy cost and the attack's success, and (2) it would be hard to limit the number of queries some other way without crippling PATE's utility, because our attack is successful while using the same number of queries used in common scenarios from the literature.

Theoretically, the attack would be mitigated if PATE returned a consistent answer for each query. PATE can thus try to cache answers to past queries and not recalculate them. Unfortunately, this defense would be exposed to adversarial perturbations that try to evade the caching mechanism without affecting predictions, and would not be possible for settings that keep queries confidential and/or include decentralized aggregation [6].

Finally, we can try to prevent sensitive information from leaking onto vote histograms. Particularly, models that generalize well across subgroups will be more immune to an attacker inferring group membership via consensus. This reduces to the problem of *subgroup fairness*, an active line of work with many proposed approaches [15]–[19] but no silver-bullet solutions.

**Limitations.**  Empirical analysis of sensitive-attribute leakage onto vote histograms (Section 2) can be expanded to improve more sophisticated attackers, other scenarios, and also other forms of sensitive information that can leak onto histograms. We instead focus our work on extracting histograms from PATE, noting that this can be used as a foundation for various different attacks.

A full optimization procedure takes a noticeably long time (roughly 10 minutes for a single step and 13 hours to convergence on a histogram), which prevented us from fully optimizing its hyper-parameter choices. This is however a limitation of our current experimental setup, not of the attack, bearing the main consequence that we are potentially under-estimating our attack's capabilities.

**Related work.**  PATE is a widely-adopted framework for differentially-private ML, with myriad applications [3], [4] and extensions [5]–[7]; our attack is generally applicable to many of those frameworks, which inherit their privacy analysis from PATE.

Another prominent decentralized ML framework, Federated Learning (FL) [20], has been extensively investigated from a privacy perspective. As we did for PATE in this work, prior work attacking FL uncovered numerous forms of leakage. For example, Hitaj et al. [21] reconstructed the average training set representation of each classes; Geiping et al. [22] reconstructed training data with high fidelity; Nasr et al. [23] mounted a membership inference attack against the clients; Wang et al.[24] showed how a malicious server could distinguish multiple properties of data simultaneously; and Melis et al. [25] inferred the clients' training data sensitive properties. These prior efforts all focus on FL, and are orthogonal to ours. We are the first to evaluate any attack against PATE.

**Conclusion**  We are the first to audit the confidentiality of PATE from an adversarial perspective. Our attack extracts histograms of votes, which can reveal attributes of the input such as race or gender, or help attackers characterize teacher partitions. The attacker's success is not highly correlated with their queries' privacy cost, which is monitored by PATE. Thus, mitigations of this attack are nontrivial and/or significantly hinder prediction utility. Particularly, using larger Gaussian noise, even when it fortifies the differential privacy guarantee, actually increases risk to the confidentiality of the vote histogram. This surprising tension demonstrates that care must be taken to analyze the protection differential privacy provides within a given threat model, rather than treat it as a silver bullet protecting against any form of leakage.

**Broader Impact**  Our work studies information leakage in a widely-adopted system, thus promoting our understanding of its risks. Our adversarial method can be used by developers and auditors to evaluate the confidentiality and privacy promises of PATE-based frameworks.
Our observation that differential privacy does not prevent but rather enables the attack is the first of its kind in that it reveals a discrepancy between differential privacy and societal norms of privacy. Characterizing this distinction is essential to building technology that uses technical definitions of privacy as an instrument to protect privacy norms.

## Acknowledgments

We would like to acknowledge our sponsors, who support our research with financial and in-kind contributions: Amazon, CIFAR through the Canada CIFAR AI Chair program, DARPA through the GARD program, Intel, Meta, Microsoft, NFRF through an Exploration grant, NSERC through the Discovery Grant, the OGS Scholarship Program, a Tier 1 Canada Research Chair and the COHESA Strategic Alliance. Resources used in preparing this research were provided, in part, by the Province of Ontario, the Government of Canada through CIFAR, and companies sponsoring the Vector Institute. We also thank members of the CleverHans Lab for their feedback.

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
