# OpenReview forum: "In Differential Privacy, There is Truth: on Vote-Histogram Leakage in Ensemble Private Learning"
_NeurIPS.cc/2022/Conference — NeurIPS 2022 Accept_

### Official Review · Reviewer_acw5 · 2022-07-08

**Rating:** 10
**Confidence:** 4
**Soundness:** 4 excellent
**Presentation:** 4 excellent
**Contribution:** 4 excellent

**Summary:**

This work provides an in-depth investigation of a particularly relevant failure mode of Differential Privacy (DP) when training ML models using the PATE framework.  In a nutshell, PATE takes the prediction of multiple independently trained models and outputs a prediction that is DP by first adding noise to the histogram and then taking the prediction with most votes.
This paper describes a privacy attack based in querying the database multiple times with the same question.  By building up a distribution of these (noisy) responses, they show it is possible to back out the underlying histogram that the noise was meant to protect.

**Questions:**

Honestly, I read everything carefully (including the appendix) and I found it quite clear, thank you!

**Limitations:**

The authors address the limitations of their work.

Not only I don't have any qualms with this work, but I believe that it is an important contribution that could have a major positive societal impact (if the work is taken seriously by the privacy community).
I think this paper should definitely be given a spotlight talk.

**Strengths And Weaknesses:**

Strengths:
- The paper is quite enjoyable to read -- it is well-written, easy to follow, logical, and concise.
- It will likely spur much-needed discussions about the practical usefulness of DP in real applications.

Weaknesses: I don't see any significant weakness.

Minor suggestions:
- Instead of using $M$ for number of samples and $m$ for number of classes, use different letters.
- (Line 263-176) I would prefer if in the appendix there was an explanation of precisely how the privacy cost is computed so that the reader doesn't need to refer to Papernot et al. as much.

Tiny typo: (line 34) *a* attacker

---

> ### Author Response · Authors · 2022-08-02
> **Response to Reviewer acw5**
>
> Thank you! We revised the manuscript to address the editorial comments and will also add a concise exposition of the privacy-cost computation. We are happy you found the manuscript both important and enjoyable, and hope to present it in the upcoming NeurIPS.

---

> > ### Comment · Reviewer_acw5 · 2022-08-09
> > **I will keep my score :)**
> >
> > I read all the reviewers and the authors' replies. I think the authors' replies were very good, and I continue to have my super positive opinion about this paper. I hope it gets accept with a spotlight talk.

---

### Official Review · Reviewer_pnDr · 2022-07-11

**Rating:** 3
**Confidence:** 3
**Soundness:** 2 fair
**Presentation:** 2 fair
**Contribution:** 2 fair

**Summary:**

This paper studies the confidentiality of PATE from an adversarial perspective. This paper shows that the attackers can recover the histograms of votes by querying the voting results multi-times. This paper shows an interesting result that adding larger Gaussian noise actually increases the risk to the confidentiality of the vote histogram.




**Questions:**

1. Does the model update frequently? Why cannot the model send the same result to the same query?

2. Why do you use the Euclidean distance to measure the distance between $Q\hat{H}$ and $\bar{q}$? Why do not use KL or Wasserstein metric to measure the distance?

3. In Line 247, what's the $\epsilon$ and $delta$ for the DP guarantee? In Line 250, what's the corresponding budget for $\sigma = 40$ etc?

4. In line 248, what's the DP budget for each query? And what's the total budget for $10^4$ queries?

5. In line 282, $(8, 10^{-5})$ is the DP budget for one query?

6 In Figure 4, what's the corresponding DP budget for $\sigma=40$?

7. The results shown in Figures 6 and 7 are counter-intuitive. What's the corresponding DP budget for each query? To verify the correctness, it's better to add a large enough noise (i.e. the standard deviation of the Gaussian noise is close to inf.). Another baseline is that we add no noise to the results. Can we reach the same conclusion?

8. Is it possible to perform some experiments that the attacker can steal the private attributes from the recovered histogram with some DP guarantee?


**Ethics Review Area:**

["I don’t know"]

**Limitations:**

See the Weaknesses and Questions part.

**Strengths And Weaknesses:**

Pros: This paper proposed an attack method that can extract the voted histograms by querying the voting results multi-times. This paper uses an example to show that sensitive attributes can be leaked from the voted histograms.

Cons: The experimental settings are not clear. More experiments are needed to support the corresponding claims.

---

> ### Author Response · Authors · 2022-08-02
> **Response to Reviewer pnDr**
>
> **1. Does the model update frequently? Why cannot the model send the same result to the same query?**
>
> We discuss the proposal to send “same result to the same query” under Mitigations (Section 6). In short, attackers can adversarially adapt to this mitigation, and it is incompatible with decentralized deployments that maintain query confidentiality.
>
> **2. Why do you use the Euclidean distance to measure the distance between $Q(\hat{H})$ and $\bar{q}$? Why do not use KL or Wasserstein metric to measure the distance?**
>
> (we are assuming the reviewer is referring to the optimization objective, where we used Euclidean distance)
> It is possible that a different distance computation would work even better. Better optimization could only improve our results. We acknowledge and justify (see Limitations) that exploring the entire space of optimization hyperparameters is left for future work.
>
> **3. In Line 247, what's the $\sigma$ and $\delta$ for the DP guarantee? In Line 250, what's the corresponding budget for $\sigma=40$ etc?**
>
> Line 247: $\delta$ is $10^{−5}$ for MNIST and  $10^{−6}$ for SVHN. $\sigma$ is 1.97 for MNIST and 4.96 for SVHN, the same as the privacy budget stated in line 247.
>
> Line 250: this is about an attacker who is not limited by a budget, but instead a hard query-number limit of $10^4$. We do report the privacy cost of each attack (see below).
>
> **4. In line 248, what's the DP budget for each query? And what's the total budget for $10^4$ queries?**
>
> The query-number-limited attacker’s privacy costs are given in Figures 6 and 7.
>
> **5. In line 282,  $(8, 10^{-5})$ is the DP budget for one query?**
>
> DP-SGD does not have a query “budget” because the guarantee it provides comes from clipping and noising gradients: a model trained with DP-SGD can answer an unbounded number of queries (the privacy cost is fixed upon completing training). PATE instead obtains privacy guarantees at inference, by noising the histogram of votes predicted by the teachers that make up the ensemble, and limiting the queries using a budget. In line 282, our intention was that the DP guarantees of the DP-SGD instantiation in [14] are equivalent to those of a PATE with a budget of 8 and a delta of $10^-5$.
>
> **6. In Figure 4, what's the corresponding DP budget for $\sigma=40$?**
>
> In Figure 4, each subfigure caption specifies the budget used and the value of $\sigma$ used. These two values are meant to be specified separately; our use of parentheses was typographically confusing (we fixed that).
>
> **7. The results shown in Figures 6 and 7 are counter-intuitive. What's the corresponding DP budget for each query? To verify the correctness, it's better to add a large enough noise (i.e. the standard deviation of the Gaussian noise is close to inf.). Another baseline is that we add no noise to the results. Can we reach the same conclusion?**
>
> We added an initial experiment with the baselines requested in Appendix F (currently includes the baselines for a single histogram, we will add the rest). We observe that when the noise is close to 0, the attacker’s error is very high. It decreases when we increase the noise, but only up to a certain point. When the noise crosses a threshold, the attacker’s error starts going up again (very moderately). This is consistent with what we expect (0 noise reveals nothing about the histogram except the argmax class, whereas “infinite” noise means that PATE’s output distribution is uniform regardless of the underlying histogram; thus, we can only hope to extract histograms when the noise level is not in either extremity).
>
> **8. Is it possible to perform some experiments that the attacker can steal the private attributes from the recovered histogram with some DP guarantee?**
>
> Yes, we added an end-to-end attack, now in Appendix E. The results in the end-to-end scenario, where the histogram is extracted from PATE and then used for minority-group membership inference, mirror the results in Section 2 where the attack is not end-to-end and the histogram is directly given to the attacker.

---

> > ### Comment · Reviewer_pnDr · 2022-08-08
> > **Thanks very much for the authors' responses.**
> >
> > Thanks very much for the authors' responses. Some of my concerns are addressed. However, I still have concerns about comment 7. If the added noise is infinite, the output distribution will be dominated by the noise. How can the attacker infer useful information from fully random data? It's better to add more descriptions about this result and sanity checks in the experiments.

---

> > > ### Author Response · Authors · 2022-08-09
> > > **Point 7**
> > >
> > > Thank you for your response. To clarify our response to point 7: the reviewer is right that if the added noise is infinite, the attacker does not learn anything useful. In Appendix F (which was added for the rebuttal) we experiment with very large and very small noise scales. The results imply that adding too much or too little noise degrades the utility of the adversary. There is a “sweet spot” for which enough noise is added to create the leakage, but not so much that the output of the noisy argmax becomes completely random.

---

> ### Author Response · Authors · 2022-08-05
> **Response to Reviewer pnDr**
>
> Dear Reviewer pnDr, it is our pleasure to discuss with you the questions you have about the paper. We are available all time.

---

### Official Review · Reviewer_42Mo · 2022-07-11

**Rating:** 7
**Confidence:** 4
**Soundness:** 2 fair
**Presentation:** 3 good
**Contribution:** 2 fair

**Summary:**

The paper first shows that the prediction histogram from several teachers will leak the membership of minority group for this test data. Then in the paper, an attack is proposed to show the possible leakage of histogram under a modern DP approach, PATE. Finally, it concludes DP doesn't protect the leakage of minority group membership.

**Questions:**

1. I am not clear why the introduced minority group membership leakage problem is a privacy problem for training data. The motivation is a bit confused.
2. It is better to check the difference between $Q(H)$ and $\bar{q}$ to support the explanation for better attack performance with higher noise levels.

**Limitations:**

The authors list some limitations of their proposed attack algorithm. However, I am concerned whether the conclusion is meaningful to the community. PATE is a DP algorithm designed for training data and it likely will not protect the test data's information, e.g. the minority group membership discussed in this paper.

**Strengths And Weaknesses:**

### Strengths
1. This paper takes effort to show what differential privacy can/cannot protect. This big topic is meaningful to the differential privacy community and the one who cares about privacy in applications.
2. The paper is written well.

### Weaknesses
1. The main conclusion of this paper is that the differential privacy doesn't protect the membership of minority group for *test data*. This conclusion seems not very surprising to me as differential privacy is to protect the privacy of *training data*.
2. The experiment result also shows that the leakage of true histogram is more severe when the privacy cost is smaller. This result is counter-intuitively and the explanation is not very convincing to me. The estimation for $q_i$ from a sequence of i.i.d. samples (0/1) from Bernoulli random variable likely wouldn't be more accurate when $q_i$ is larger, but depends on its variance $q_i(1-q_i)$. I would suggest authors to further check the difference between $Q(H)$ and $\bar{q}$ and see if this difference is smaller when the injected noise is larger.
3. The experiment doesn't include the end-to-end evaluation between privacy cost to minority group membership leakage.

--------After Rebuttal--------

Thank the authors for their reply, which solves W2 and W3 and alleviates my concern in W1.

---

> ### Author Response · Authors · 2022-08-02
> **Response to Reviewer 42Mo**
>
> **The main conclusion of this paper is that the differential privacy doesn't protect the membership of minority group for test data. This conclusion seems not very surprising to me as differential privacy is to protect the privacy of training data.**
>
> First, while it is not surprising that DP for training data does not protect test data, it is less obvious that it actually causes leakage of extremely sensitive attributes such as minority-group membership. While this leakage is not in DP’s threat model, it undermines the societal norms that DP is meant to uphold. Ours is the first work to illustrate how adding DP noise can cause leakage (of any kind).
>
> Second, it would be incorrect to say that our attack can only infer test-set information. Histograms contain ample information about the training data, as well, because different training sets will present different histograms and consensus levels across test instances (e.g., our attacker can detect biased training sets where consensus varies across populations).
>
> **The experiment result also shows that the leakage of true histogram is more severe when the privacy cost is smaller. This result is counter-intuitively and the explanation is not very convincing to me. The estimation for $q_i$ from a sequence of i.i.d. samples (0/1) from Bernoulli random variable likely wouldn't be more accurate when $q_i$ is larger, but depends on its variance . I would suggest authors to further check the difference between $Q(H)$ and $\bar{q}$ and see if this difference is smaller when the injected noise is larger.**
>
> The reviewer touches on an interesting subtlety. It is absolutely true that $\bar{q}_i$ (the MC estimator) will be less accurate as more noise is added. This conclusion is as theoretically sound as it is intuitive. However, the degree in which {$\bar{q}_i$}$_n$ (collection of estimators) reveal information about {$H_i$}$_n$ (the vote histogram itself, which is our real target) does sometimes drop as more noise is added. To illustrate this, consider the case of $\sigma=0$, i.e. no noise is added at all. PATE will always return the histogram’s argmax class. Regardless of how many queries are used, our adversary can never learn anything about the histogram except for the identity of the argmax class (which is already known after a single query). In terms of $\bar{q}_i$, the estimator of the argmax class will be 1, and the others 0, which is actually an entirely precise estimation $(\bar{q}==Q(H))$ yet it reveals very little about the vote histogram. Conversely, when sufficient noise is added and given enough queries, the entire vote histogram can be estimated with very high accuracy.
>
> Please see also the experiment we added in Appendix F. It shows that when the noise is close to 0, the attacker’s error in estimating $H$ is very high. It decreases when we increase the noise, but only up to a certain point. When the noise crosses a threshold, the attacker’s error starts going up again (very moderately). This is consistent with what we expect: 0 noise reveals nothing about the histogram except the argmax class, whereas “infinite” noise means that PATE’s output distribution is uniform regardless of the underlying histogram; thus, we can only hope to extract histograms when the noise level is not in either extremity.
>
> **The experiment doesn't include the end-to-end evaluation between privacy cost to minority group membership leakage.**
>
> We added an end-to-end attack, now in Appendix E. The results in the end-to-end scenario, where the histogram is extracted from PATE with a privacy budget equal to 1.9 and then used for minority-group membership inference, mirror the results in Section 2 where the attack is not end-to-end and the historgram is directly given to the attacker.

---

> > ### Comment · Reviewer_42Mo · 2022-08-04
> > **Response**
> >
> > Thanks for authors' clarification and additional experiments, which solve my concern in W2 and W3 and alleviate my concern for W1. I will raise my score to 7.
> >
> > I have one more comment for the W1. The threat proposed in this paper is to infer whether a given data is belong to minority group in the training distribution. Differential privacy is designed to reveal the distribution-level property, while to protect any single data in the training set. Therefore, it is intuitive that differential privacy doesn't provide the protection to the proposed attack.

---

> > > ### Author Response · Authors · 2022-08-05
> > > **Response to Reviewer 42Mo**
> > >
> > > Thank you so much for your insightful feedback!

---

### Official Review · Reviewer_r8cf · 2022-07-11

**Rating:** 4
**Confidence:** 3
**Soundness:** 3 good
**Presentation:** 3 good
**Contribution:** 2 fair

**Summary:**

This paper reveals that the noise added to the PATE voting mechanism (to attain a differential privacy guarantee) enables new forms of leakage of sensitive information. A simple adversary is enough to exploit this noise to extract high-fidelity histograms of the votes. As pointed out in this paper, a low consensus vote indeed indicates minority-group membership. The main technical contribution is the model to extract PATE histograms.


**Questions:**

 Weaknesses: The technical part seems too simple. I'm not sure such a simple attack model contains enough technical contribution to appear in NeurIPS. This paper might be more appropriate to be published (as a report paper) at a security conference.

**Limitations:**

 Weaknesses: The technical part seems too simple. I'm not sure such a simple attack model contains enough technical contribution to appear in NeurIPS. This paper might be more appropriate to be published (as a report paper) at a security conference.

**Strengths And Weaknesses:**

Strengths: This paper is well written. The proposed attack model is simple but interesting. I believe the authors' discovery is significant.

---

> ### Author Response · Authors · 2022-08-02
> **Response to Reviewer r8cf**
>
> Thank you for the feedback. First, while we made a deliberate effort to explain our attack and results in the simplest of terms, they include a full stack of technical contributions and conceptual insights, including
> 1. Introduction of a novel attacker model, not considered before for PATE, and an accurate formulation of the attacker’s problem.
> 2. The insight that histograms are sensitive information, and the observation that DP mechanisms can actually cause leakage of sensitive information.
> 3. A carefully-devised optimization objective that required finding a closed-form expression and an approximation formula.
> 4. An experimental design that adequately controls all PATE and DP parameters while offering clear and useful reporting metrics.
>
> Second, we would like to note that the intuitive high-level structure of the attack itself (Monte Carlo estimation followed by an optimization procedure) is an advantage for attackers. We believe this only reinforces the impact of our conclusions.
>
> We are happy to further discuss any questions or concerns the reviewer has regarding specific elements of our work such as the optimization procedure or experimental design.

---

> ### Author Response · Authors · 2022-08-05
> **Response to Reviewer r8cf**
>
> Dear Reviewer r8cf, it is our pleasure to discuss with you the questions you have about the paper. We are available all time.

---

### Meta-Review · Area_Chair_8esd · 2022-08-26

**Recommendation:** Accept
**Confidence:** Less certain

**Metareview:**

Although the fact that DP does not protect against population statistics is a widely known fact, the paper weaves this together with PATE (which relies on DP statistics) to demonstrate the danger of mis interpreting the protection guarantees provided by DP. This is a point worth discussing among the privacy and security community.

**Award:**

No

---

### Decision · Program_Chairs · 2022-09-14

Accept